# Opt-out rates and reasons for non-participation in a single-arm feasibility trial (ENGAGE) of a guided internet-administered CBT-based intervention for parents of children treated for cancer: a nested cross-sectional survey

Josefin Hagström,[1] Joanne Woodford [ID],[1] Agnes von Essen,[1] Päivi Lähteenmäki,[2,3] Louise von Essen[1]

JH and JW are joint first authors.

For numbered affiliations see end of article.

**Correspondence to**
Professor Louise von Essen;
louise-von.essen@kbh.uu.se

## ABSTRACT

**Objectives** Difficulties with recruitment into clinical trials are common. An opt-out recruitment strategy, whereby potential participants can decline further contact about a study (opt-out), and non-responders are contacted, may facilitate participation. Primary objectives examined opt-out and consent rates, mode and time point of opt-out, and sociodemographic characteristics of those who opted out versus those who chose to participate in a single-arm feasibility trial (ENGAGE) of a guided, internet-administered, cognitive–behavioural therapy-based intervention for parents of children treated for cancer. Secondary objectives examined reasons for non-participation.

**Design** A cross-sectional survey nested within the ENGAGE feasibility trial.

**Setting** The intervention was delivered from Uppsala University, with parents located throughout Sweden.

**Participants** Potential participants were recruited 3 months–5 years following their child ending treatment for cancer and were identified via their personal identification number (via the Swedish Childhood Cancer Registry and Swedish Tax Agency) and invited via postal invitation packs and could opt out via post, online, telephone or email. Those who did not opt out or consent, within 4 weeks, received up to five telephone calls and/or one postal reminder.

**Results** Of 509 invited, 164 (32.2%) opted out, 78 (47.6%) via post, 53 (32.3%) via telephone, 24 (14.6%) online, and 6 (3.7%) via email, 88 (53.7%) opted out after at least one telephone call and/or postal reminder. There was a trend for parents with lower educational levels to opt out. No need of psychological support, lack of time, and no interest in internet-administered self-help were frequently reported reasons for non-participation.

**Conclusions** Results emphasise the importance of using different opt-out modes and suggest future research should consider how to widen study participation for parents with lower education levels. Self-identifying a need for psychological support and the acceptability of internet-administered self-help are important factors for

### Strengths and limitations of this study

► Parents invited are a nationally representative sample of parents of children treated for cancer.
► The use of different modes of opt-out (eg, post, online, telephone, or email) has rarely been investigated.
► The use of closed and open questions allows for a broad understanding of reasons for non-participation which can inform strategies to increase recruitment rates of future clinical trials.
► The study was not designed to compare recruitment rates across different strategies and conclusions concerning the effectiveness of opt-out versus opt-in strategies cannot be drawn.
► We did not collect sociodemographic variables of non-responders and therefore it was not possible to examine whether the sample of parents who opted out, or chose to participate, is representative of the overall population in relation to key sociodemographic variables.

participation and should be considered in future research to increase recruitment.

**Trial registration number** ISRCTN57233429.

## INTRODUCTION

### Background

Difficulties with recruitment into clinical trials are common,[1] significantly contributing to research waste.[2] Insufficient recruitment rates have been cited as the leading cause of clinical trials being terminated[3] and reviews have found only 56% of publicly funded clinical trials reach their target sample size.[4] Difficulties with recruitment into clinical trials may result in delays to the generation of robust evidence to inform decisions to

adopt interventions into routine healthcare.[5] While there is an increased focus on identifying effective clinical trial recruitment strategies,[6] current evidence is limited[7] with little high-quality evidence to inform decisions concerning the use of different recruitment strategies in clinical trials.[6]

The ENGAGE feasibility trial[8 9] is a single-arm feasibility trial of a guided, internet-administered low-intensity cognitive-behavioural therapy (LICBT)-based intervention, designed to target common mental health difficulties (eg, depression and generalised anxiety disorder) in parents of children treated for cancer.[10] The ENGAGE feasibility trial examines clinical (eg, intervention drop-out, adherence, and acceptability) and procedural (eg, recruitment and retention) uncertainties. Examining recruitment uncertainties is important given that recruitment of participants into mental health intervention trials is particularly challenging.[11] Barriers may include a reluctance to seek help,[12] stigma,[13 14] and low perceived need for support.[15] Recruitment into clinical trials of e-mental health (e-MH) interventions (eg, internet-administered CBT) has been identified as particularly challenging,[16] with barriers including lack of trust, poor digital health literacy, low confidence in using technology, low acceptability and preferences for face-to-face interventions.[17–20]

Given challenges with recruitment into e-MH intervention trials, there is a need to examine the effectiveness of different recruitment strategies to increase recruitment rates. One strategy associated with increased recruitment rates and less biased samples is the opt-out strategy.[21] Potential participants invited into a study are provided with a method of actively declining contact with the research team (opting out) if they do not wish to participate in or receive additional study information. Those who do not opt out are contacted by the research team, for example, via the telephone, to provide more study information. Contacting non-responders may result in increased recruitment rates in comparison with opt-in strategies, whereby potential participants must take action to signal willingness to participate to the research team.[1 7 11 22] An opt-out strategy may be particularly effective for e-MH intervention trials.[16] Mental health difficulties such as depression include lack of motivation, concentration difficulties, lack of belief in deserving intervention and negative expectations concerning recovery, which have been found to be barriers to help-seeking.[16 23 24] An opt-out strategy may facilitate participation, for example, potential participants experiencing low levels of motivation or problems with self-initiating behaviour[25] are contacted by the research team and provided with an opportunity to receive more information. Indeed, opt-out strategies have been found to result in the recruitment of populations who are less healthy and more functionally impaired than opt-in strategies.[21] While successfully used with adult and paediatric populations,[26] to the best of our knowledge, an opt-out strategy has not been used with parents of children treated for cancer.

Furthermore, literature is scarce concerning reasons for non-participation reported by parents of children treated for cancer. Wider literature suggests common reasons include lack of perceived personal benefit, time and energy, and preconceptions about research.[27 28] While qualitative research with parents of children treated for cancer has been undertaken to inform the development of recruitment strategies for clinical trials,[29] to the best of our knowledge, the population's reasons for non-participation have not been investigated. Understanding reasons for non-participation may enable the identification of modifiable barriers to participation (eg, treatment preferences, burden of trial procedures)[14] and inform the development of recruitment strategies to help overcome these barriers.[14]

### Aims and objectives
Primary objectives examined: (1) opt-out and consent rates; (2) mode of opt-out; (3) time point of opt-out; and (4) sociodemographic characteristics of those who opted out versus those who chose to participate in the ENGAGE feasibility trial.[8 9] Secondary objectives examined reasons for non-participation.

### METHODS AND ANALYSIS
### Study design
A cross-sectional survey with closed and open questions examining opt-out rates and reasons for non-participation nested within the ENGAGE feasibility trial[8 9] reported, where applicable, in line with the Checklist for Reporting of Survey Studies.[30]

### Participants
Eligible participants were: (1) parents of a child diagnosed with cancer during childhood (0–18 years) who had completed cancer treatment 3 months–5 years previously; (2) living in Sweden; (3) able to read and write Swedish; (4) able to access email, the internet, and a mobile telephone and BankID; and (5) self-reporting a need for psychological support related to the child's cancer disease and treatment and a wish to work with the intervention. Parents were excluded if they: (1) had a self-reported or psychologist-assessed (based on the Mini-International Neuropsychiatric Interview[31]) severe and enduring mental health difficulty; and/or misuse of alcohol, street drugs, and/or prescription medication; (2) were acutely suicidal; and/or (3) attended psychological treatment.

### Recruitment
Potential participants were recruited using two strategies: (1) postal study invitation packs; and (2) online advertisements via cancer organisations and interest groups. In this publication, recruitment procedures and data associated with postal study invitation packs are reported.

Personal identification numbers of children who completed cancer treatment 3 months–5 years previously

were obtained from the Swedish Childhood Cancer Registry and linked to the names and addresses of their parent(s) via NAVET, a population registry held by the Swedish Tax Agency. Parents were invited in blocks of 100 until the target sample size (N=50) was reached. The first block invited was preselected with parents of children treated for cancer who ended treatment near to 5 years ago. The following four blocks were randomly selected. A de-identified list of all parents provided by NAVET was prepared by a research team member. To ensure allocation concealment, a member of the U-CARE web development team, external to the research team, used a computer-generated simple randomisation procedure to allocate parents into recruitment blocks. Postal study invitation packs were sent to parents' home addresses and included: (1) a study invitation letter, signed by the principle investigator (LvE) and a parent research partner (PRP); (2) a study information sheet and link to a secure website, the U-CARE-portal (Portal, an in-house platform which hosted the study and is designed to deliver internet-administered psychological interventions and support the collection of research data[32 33]); (3) a paper reply slip to register interest in participation; (4) an opt-out form and reasons for non-participation survey; and (5) a freepost envelope. The ENGAGE feasibility trial also included a Study Within A Trial investigating the effect of personalised versus non-personalised study invitation letters on recruitment and retention,[9] and results have been published separately.[34]

Parents could access study information, in text and video format, and provide informed consent via the Portal. Parents who wished to opt out could do so by completing an opt-out form and reasons for non-participation survey via the Portal or by paper via the post (see the Outcomes section). Parents could also opt out by telephoning or emailing the research team.

### Procedure
#### Opt-out and reminder telephone calls
Parents could sign an online form via the Portal using BankID, a secure electronic citizen authentication system used in Sweden, and were presented with a short reasons for non-participation survey (see the Outcomes section). Alternatively, parents could complete a paper-based survey, provide their name and ink signature, and return via the post. Parents who opted out via telephone or email were asked to optionally provide a reason for non-participation.

Parents who did not respond (eg, opt out or provide consent) within 4 weeks upon the first postal study invitation pack, received up to five telephone reminder calls over 4 weeks. Telephone numbers were identified using internet search engines. A maximum of five reminder calls were made over 4 weeks, intended to confirm receipt of the postal study invitation and provide an opportunity to ask questions, or opt out from the study and further contact. Voicemails were left if the voicemail inbox had a personal recording clearly stating the

parent's name. Communication between the parent and research team was documented in a recruitment case report form (CRF). If a telephone number could not be identified, a study invitation reminder letter and study invitation pack were resent. Personal details of parents who opted out were deleted from the research database.

### Informed consent, screening and baseline
Parents who chose to participate provided informed consent via the Portal and were contacted by the research team to schedule an eligibility interview with a licensed psychologist over the telephone. All parents were required to complete an online Portal assessment at baseline, or telephone if preferred. Full study procedures can be found in the ENGAGE feasibility trial protocol.[8]

### Intervention
The intervention is an internet-administered, guided, LICBT-based intervention (the EJDeR intervention (intErnetbaserad sJälvhjälp för förälDrar till barn som avslutat en behandling mot canceR)).[10] Following phase I of the Medical Research Council's complex interventions framework,[35] development of EJDeR was informed by a systematic review,[36] qualitative interview studies,[37 38] a single-arm trial,[39] participatory action research[40] and a cross-sectional online survey.[41] A PRP group of two mothers and two fathers of a child treated for cancer informed further refinements.[10] EJDeR is delivered on the Portal, and designed to be worked with over 12 weeks and consists of four modules: (1) Introduction and Psychoeducation; (2) Behavioural Activation; (3) Worry Management and (4) Relapse Prevention.[10] Parents are guided by e-therapists, including an initial assessment session via a secure inbuilt video conferencing system or telephone call, weekly written messages via the Portal, and a 'booster' session mid-way through the intervention, via video conferencing or telephone call.

### Outcomes
An online and paper-based survey, consisting of 13 items, in Swedish. The survey comprises two subsections: (1) sociodemographic characteristics (four items) and (2) reasons for non-participation (nine items) . An open question at the end of the questionnaire offered an opportunity for parents to provide any reasons for non-participation.

### Sociodemographic characteristics
Background and sociodemographic characteristics were collected from parents who opted out via the online or paper-based survey: (1) gender (male/female/other); (2) age (years); (3a) relationship status (yes/no); (3b) cohabitation status (yes/no) and (4) highest level of education (free-text). Parents opting out via the telephone or email were not asked to provide sociodemographic characteristics.

## Reasons for non-participation

Parents were asked a closed, multiple-choice question regarding reasons for non-participation, reasons were informed by previous research,[42 43] and parents could select all applicable reasons: (1) lack of time; (2) physical health; (3) not wanting to participate in research; (4) not liking to talk about personal problems; (5) too tired; (6) not interested in internet-administered self-help; (7) no need for psychological support; (8) child's health condition and (9) child has recently passed away. The open question at the end of the reasons for non-participation questionnaire was: 'If you do not want to participate in the project for reasons other than those mentioned above, please write these below'.

Whereby the parent opted out via telephone or email, a CRF was completed by the research team. The CRF included the reason(s) for non-participation listed in the survey and a free-text box in response to the open question.

## Sample size

No formal sample size calculation was made[44] however recommendations for assessing feasibility outcomes were followed[45] with an aim to recruit 50 parents. It was estimated approximately 600 parents would be invited to meet the sample size.[9]

## Progression criteria

A priori progression criteria[46] were set, these can be found in the published protocol for the ENGAGE feasibility trial.[8] A criterion that ≥9% of the total population invited into the study via the Swedish Childhood Cancer Registry chose to participate was set.

## Data analysis

Data analysis was performed in Excel and SPSS for Windows, V.27.0 (IBM Corporation). Recruitment flow via postal study invitation packs is illustrated using an adapted Consolidated Standards of Reporting Trials flow diagram.[47] Descriptive statistics were used to describe numbers and proportions who opted out; opted out via each mode (postal/online/telephone/email) and at each time point (before/after telephone and/or postal reminder/s); and responded to a multiple-choice question regarding reasons for non-participation. Study sample sociodemographic characteristics are described using descriptive statistics. $X^2$ test was used to examine differences regarding categorical variables, and t-tests were calculated for continuous variables.

An inductive qualitative content analysis[48] managed using Microsoft Excel and Word was adopted to analyse responses to the open question in the reasons for non-participation survey. Free-text responses were read by two authors (JH and AvE) to gain an overall understanding of content. Next, condensed meaning units were identified (eg, text sharing a common meaning) and assigned a code by JH and AvE separately. JH reviewed initial meaning units and codes to identify any differences in coding, with

differences discussed by JH, AvE and a third author (JW) if required. Codes were sorted into categories and subcategories by JH and AvE separately. As categories were identified, some content was recognised as not relating to reasons for non-participation, but included important content not to be discarded, for example, gratitude for the research. Categories were subsequently classified into two themes: (1) reasons for non-participation and (2) content not related to reasons for non-participation. JH, AvE and JW met regularly to discuss categories and subcategories and ensure content was mutually exclusive. All responses were subsequently reread by JH to check agreement with identified categories and subcategories. JH and AvE jointly prepared brief descriptions for categories and subcategories. Trustworthiness was established using peer examination of a supervising researcher (JW), two coders (JH and AvE) working independently, and record keeping.[49]

Other reasons for non-participation provided by parents via telephone or email to the research team and recorded on the recruitment CRF were also categorised by JH and AvE separately, with all categorisations discussed with JW to obtain consensus.

## Patient and public involvement

The PRP group (two mothers and two fathers) provided feedback on the design and wording of the study invitation letter, included within the study invitation pack. PRPs were sent a study invitation letter draft via email and asked to comment on wording, content, format, and design, and whether to include a PRP's name and signature on the letter alongside that of the principle investigator (LvE). Parents did not suggest any changes and considered the letter to be concise, validating and informative, and agreed the name and signature of a PRP should be included.[9] Patient and public involvement may have been improved by engaging PRPs in drafting the opt-out survey, for example, informing the options available in the multiple-choice question regarding reasons for non-participation.

## RESULTS

### Opt-out and consent rates

Recruitment via postal study invitation packs into the ENGAGE feasibility trial took place between 3 July 2020 and 30 November 2020. Five hundred nine parents were identified and invited. A total of 32.2% (164 of 509) opted out, and 11.8% (60 of 509) provided consent, and 11.0% were eligible and enrolled in the trial, exceeding the a priori recruitment rate progression criteria of ≥9% invited via the Swedish Childhood Cancer Registry.[8 46] Eleven per cent (56 of 509) were found to be eligible for the study.

Three parents were initially registered as opting out via the telephone by a partner who had also received a study invitation pack. Subsequently, the opt-out protocol was adapted and opting out via a partner was no longer an

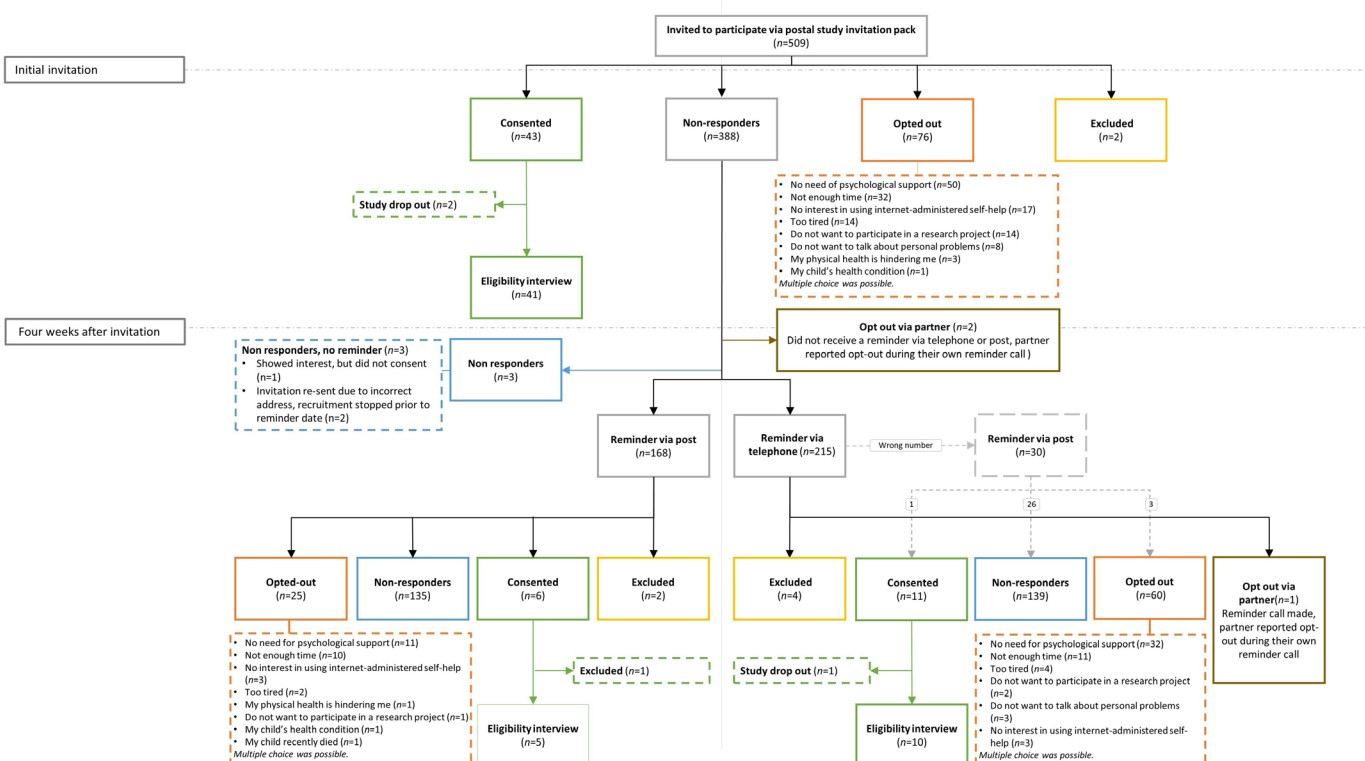

**Figure 1** Recruitment flow via postal study invitation packs.

option. Recruitment flow via postal study invitation packs can be seen in figure 1.

## Mode of opting out

The most frequent mode of opt-out was via the post (78 of 164, 47.6 %), followed by telephone (53 of 164, 32.3%), the Portal (24 of 164, 14.6%) and email (6 of 164, 3.7%). Three parents (3 of 164, 1.8%) opted out via their partner over the telephone. Two who initially opted out via telephone, subsequently opted out via the post, resulting in a total of n=80 opt-outs via the post.

## Time point of opting out

The majority of opt-outs occurred after one or more telephone reminder calls and/or postal reminders (88 of 164, 53.7%). The mean number of days between a postal study invitation pack being sent and an opt-out was 30.0 days (SD=20.6, range=2–158).

## Sociodemographic characteristics of parents who opted out versus those who chose to participate

Sociodemographic characteristics of parents (n=105) who opted out and provided sociodemographic data and parents (n=56) who consented and were screened for eligibility are shown in table 1. No differences between groups were found, though there was a trend for parents with lower educational levels to opt out, and parents with a tertiary education to choose to participate.

## Reasons for non-participation
### Closed question

A total of 83.5% (137 of 164) responded to the closed question about reasons for non-participation. The most frequently reported reason was not experiencing any need for psychological support (93 of 137, 67.9%) followed by not having enough time (53 of 137, 38.7%), and not interested in using internet-administered self-help (23 of 137, 16.8%). Responses are shown in table 2 and presented separately for female and male gender where data were available.

### Open question

Thirty parents responded to the open question regarding reasons for non-participation in the paper-based or online survey. Twenty-one provided a response via telephone or email recorded in the CRF. Content analysis of responses within the theme Reasons for non-participation resulted in six categories: Timing, Emotional well-being, Support, Perceived ineligibility, Other commitments, and Reluctance to engage in internet-administered self-help. Responses within the theme Content not related to reasons for non-participation were categorised as: Not for me, but…. and When my child was ill. Themes, categories, subcategories and quotations are shown in table 3.

### Reasons for non-participation

The category Timing includes the subcategories: *Earlier need* and *Not right now*. *Earlier need* concerns the invitation coming 'too late', that is, after treatment completion, with

**Table 1** Sociodemographic characteristics of parents who opted out and who consented and were screened for eligibility

| Characteristic | Opted out (n=105)* | Consented and screened for eligibility (n=56) | t value, χ2 value | P value |
|---|---|---|---|---|
| Age, mean (SD, range) | 44.8 (7.6, 29–68) | 43.5 (7.2, 32–62) | −0.988 | 0.325 |
| Missing (n, %) | 19 (18.1) | 0 (0) | | |
| **Gender (n, %)** | | | | |
| Female | 54 (51.4) | 33 (58.9) | 0.001 | 0.978 |
| Male | 38 (36.2) | 23 (41.1) | | |
| Missing | 13 (12.4) | 0 (0) | | |
| Relationship status (n, %) | | | | |
| Partner | 86 (81.9) | 47 (83.9) | 1.996 | 0.158 |
| Single | 8 (7.6) | 9 (16.1) | | |
| Missing | 11 (10.5) | 0 (0) | | |
| **If partner, cohabiting** | | | | |
| Yes | 78 (90.7) | 46 (97.9) | 1.035 | 0.309 |
| No | 5 (5.8) | 1 (2.1) | | |
| Missing | 3 (3.5) | 0 (0) | | |
| Level of education (n, %)† | | | | |
| Lower secondary | 6 (5.7) | 1 (1.8) | 6.085 | 0.298 |
| Upper secondary | 30 (28.6) | 11 (19.6) | | |
| Post-secondary non-tertiary | 5 (4.8) | 3 (5.4) | | |
| Tertiary education | 50 (47.6) | 40 (71.4) | | |
| PhD | 2 (1.9) | 1 (1.8) | | |
| Not possible to categorise | 1 (1.0) | 0 (0) | | |
| Missing | 11 (10.5) | 0 (0) | | |

*One hundred and five parents who opted out provided some sociodemographic data. Sociodemographic characteristics were collected from parents who opted out by paper via the post (n=80) or Portal (n=24). One parent opted out via email and voluntarily provided some sociodemographic data.
†Level of education was self-reported as a free-text response, with responses categorised by the research team.

parents positing hypothetical participation had they been approached earlier. *Not right now* concerns not currently interested in participation, however potential interest another time.

The category Emotional well-being includes the subcategories: *Coping strategies, Family well-being, Fatigue* and *No need. Coping strategies* concerns use of coping strategies. Parents stated wanting to forget 'what has been' (eg, the

**Table 2** Reasons for non-participation in response to the closed multiple-choice question

| Reasons | Female (n=50*) n | Female (n=50*) % | Male (n=35*) n | Male (n=35*) % | Total (n=137) n | Total (n=137) % |
|---|---|---|---|---|---|---|
| I do not experience any need for psychological support | 32 | 64.0 | 27 | 77.1 | 93 | 67.9 |
| I do not have enough time | 24 | 48.0 | 19 | 54.3 | 53 | 38.7 |
| I am not interested in using internet-administered self-help | 10 | 20.0 | 8 | 22.9 | 23 | 16.8 |
| I am too tired to participate in a research project | 9 | 18.0 | 6 | 17.1 | 20 | 14.6 |
| I do not want to participate in a research project | 7 | 14.0 | 8 | 22.9 | 17 | 12.4 |
| I do not like talking about my personal problems | 5 | 10.0 | 4 | 11.4 | 11 | 8.0 |
| My physical health is hindering me | 3 | 6.0 | 1 | 2.9 | 4 | 2.9 |
| I do not want to participate because of my child's health condition | 2 | 4.0 | 0 | 0 | 2 | 1.5 |
| I do not want to participate because my child recently died | 1 | 2.0 | 0 | 0 | 1 | 0.7 |

*n=85 parents who provided data regarding gender (female/male).

**Table 3** Themes, categories, subcategories and quotations

| Theme | Category | N (n=51) | Subcategory | Quotation (P: reported by parent; R: recorded by researcher) |
|---|---|---|---|---|
| Reasons for non-participation | Timing | 20 | Earlier need | 'We would have needed help a long time ago, but we have managed well and right now we are doing well' (P) <br> 'The need was bigger right after completed treatment' (R) |
| | | | Not right now | 'Could be interesting another time' (P) |
| | Emotional well-being | 16 | Coping strategies | 'We/I don't have the energy to be involved in sickness anymore, more than I am' (P) <br> 'Sure, it was heavy and tough, but life is what you make it' (P) |
| | | | Family well-being | '… my son is well today, and us parents as well' (P) <br> 'Child is doing fine' (R) |
| | | | Fatigue | 'We are tired and it's always a lot, but it's not related to the cancer [the child's]' (P) |
| | | | No need | 'Feeling no need for help, feeling good' (P) <br> 'Felt no need for therapy' (R) |
| | Support | 13 | Professional support | 'I have processed a lot on my 'own' with the help from a psychologist' (P) <br> 'Has received psychological treatment' (R) |
| | | | Social support | '[I've come a long way] with help from […] and close ones' (P) <br> 'Had a good social network helping the family' (R) |
| | Perceived ineligibility | 11 | N/A | 'Cannot read Swedish' (P) <br> 'In psychological treatment' (R) <br> 'I'm too broken in many ways, and unfortunately I cannot participate' (P) <br> 'Did not feel suitable for the project' (R) |
| | Other commitments | 7 | N/A | 'Unfortunately I don't have the time to participate' (P) <br> 'Too busy to participate' (R) |
| | Reluctance to engage in internet-administered self-help | 2 | N/A | 'Did not feel comfortable with the internet' (R) |
| Content not related to reasons for non-participation | Not for me, but … | 19 | Gratitude | 'Great that you are doing this' (P) <br> 'Very positive about project' (R) |
| | | | Need among others | 'I hope many participate, there is really a need for support after treatment' (P) <br> 'Would love to help other parents' (R) |
| | When my child was ill | 10 | Available care | 'The care after completing treatment was non-existent' (R) |
| | | | Cancer illness | 'My son was critically ill when the intestine broke' (P) |
| | | | Family life | 'When [child] was diagnosed with cancer, my husband and I tried to keep things as 'normal' as possible' (P) |

Categories are presented in order of number of included responses, whereas subcategories are presented alphabetically.
N/A, not applicable.

cancer experience) and a reluctance to 'go back and think about these things' or to remain involved with sickness. Conversely, parents reported focusing on the positive, accepting their situation, holding a 'positive' attitude towards difficult situations and post-traumatic growth. Parents could describe distancing from the cancer experience in the past and present as well as a want to 'focus on the positive' in the future. *Family well-being* concerns the entire family, for example, as their child was in good health, psychological support for the parent was not considered needed. *Fatigue* concerns being too tired to

participate and *No need* concerns parents feeling well and not experiencing a need for psychological support.

The category Prior support includes the subcategories: *Professional support* and *Social support*. *Professional support* concerns receipt of current or past support from a health professional, for example, a psychologist or counsellor, whereas *Social support* concerns support and help by family, friends and pets.

The category Perceived ineligibility concerns parents not considering themselves eligible for the study, for example, not fulfilling the inclusion criteria, for example,

insufficient knowledge of Swedish, undergoing psychological treatment as well as other reasons such as being 'too broken' to participate.

The category Other commitments concerns life commitments occupying parents' time, reference was also made to too many invitations regarding participation in research and a wish to have free time.

The category Reluctance to engage in an internet-administered intervention included responses whereby parents reported being uncomfortable with the internet or preferring face-to-face treatment.

### Content not related to reasons for non-participation

The category Not for me, but… includes the subcategories: *Gratitude* and *Need among others*. *Gratitude* concerns the importance of the study and gratitude that it was carried out. *Need among others* concerns a hope that other parents would participate and receive help, and that there is a need for support for parents after their child's cancer treatment.

The category When my child was ill includes the subcategories: *Available care*, *Cancer illness* and *Family life*. *Available care* concerns the care received during and after the child's illness and treatment. Parents mentioned there had been little care available for their well-being after their child had completed treatment. *Cancer illness* concerns details about the child's illness and treatment, for example, the duration of the treatment or the child having been ill since birth. *Family life* concerns parents describing experiences during their child's illness and treatment, for example, striving for normality or difficult family events such as separating from a partner.

### DISCUSSION

Summarising the main findings, the opt-out rate was 32.2%, the consent rate via postal study invitation packs was 11.8%, and 11.0% were eligible and enrolled in the trial. The most frequent mode of opt-out was by paper via the post (47.6%). The majority (53.7%) opted out of further contact with the research team after one or more reminders via telephone and/or post. Although not significant, there was a trend for parents with lower educational levels to opt out, and parents with a tertiary education choosing to participate. Common reasons for non-participation in response to the closed question were not experiencing any need for psychological support, not having enough time, and not having an interest in using internet-administered self-help. Content analysis of responses to the open question agreed with results from the closed question, with reasons for non-participation including emotional well-being, receipt of psychological support, and commitments limiting time to participate. Additional reasons were suggested, including study invitation timing and parents not perceiving themselves as eligible.

The overall consent rate exceeded the recruitment target outlined in study progression criteria.[8 46] Use of an opt-out strategy has demonstrated higher recruitment rates when compared with opt-in strategies.[7 50 51] For example, randomised controlled trials (RCTs) comparing opt-out versus opt-in strategies have found recruitment rates of 40%–50% for opt-out vs 26%–38% for opt-in.[21 52] However, studies using opt-out strategies are highly heterogeneous, with variations in recruitment rates, populations, designs, and interventions,[50] and few RCTs have compared opt-out versus opt-in strategies,[7 51] limiting conclusions regarding effectiveness. The ENGAGE feasibility trial did not include an opt-in comparison arm, making it impossible to determine whether successful recruitment was a result of an opt-out strategy.

Recruitment took place during the COVID-19 pandemic. Given physical distancing recommendations in Sweden, as in many countries globally,[53] people have spent increased time at home. Research has observed increased parental stress associated with physical distancing measures and the closure of schools and child-care facilities.[54] A retrospective online survey conducted in the USA found that 71.1% of parents reported increases in parenting-specific stress from before COVID-19.[54] Further, a survey conducted in the UK indicated that 85% of parents of children with cancer were worried about the virus, for example, risk of infection, reduced healthcare provision, and increased risk of social isolation, and 70% no longer deemed the hospital a safe place.[55] The context of a pandemic may have facilitated recruitment, for example, parents may have been experiencing elevated levels of fear and anxiety. Conversely, the pandemic may have been a barrier to participation for some, given that parents of children with cancer commonly focus on their child's needs, at the expense of their own well-being.[36–38]

The overall opt-out rate in the present study is higher than a 19% opt-out rate found in a longitudinal cohort pilot study of patients with angina[21] and a 21% opt-out rate seen in a survey study regarding end-of-life care.[52] However, existing literature on opt-out strategies tends to be on observational studies, as opposed to intervention or health service-orientated research[50] making it difficult to compare results with the wider literature. When comparing the opt-out rate in the present study with non-participation rates in intervention studies for parents or informal caregivers using opt-in strategies, findings are also varied. In an RCT of an intervention for parents of children with cancer to reduce exposure to tobacco smoke, 24% approached about study participation within a clinic setting declined participation[56] and in a pilot RCT of guided CBT self-help for informal caregivers of stroke survivors, 15% invited by post actively declined participation.[57] Further, in an RCT of a psychological intervention for patients with cancer in palliative care and their family caregivers, 65.9% of potential participants actively declined participation.[58]

To the best of our knowledge, this is the first study to specifically examine the rate of opt-out across four different modes of opt-out. Commonly, studies using opt-out recruitment strategies include only one mode

of opt-out, for example, via the post[26 59] or telephone.[60] The most frequent mode of opt-out in the ENGAGE feasibility trial was paper via the post (47.6 %), followed by telephone (32.3%), with findings similar to a quantitative telephone survey with military veterans examining rural and urban differences in attitudes towards mental healthcare and influence on mental health service use, whereby opt-out via the post was more frequent (75%) than opt-out via the telephone (25%).[50] Interestingly, only 14.6% of parents opted out online via the Portal in the ENGAGE feasibility trial. In comparison with the general adult population, parents of a child treated for cancer 3 months–5 years following end of treatment may be assumed to be younger, with good digital knowledge and skills.[61] Therefore, it may have been anticipated that parents would prefer opt-out online. However, systematic reviews and meta-analyses have shown postal survey response to be, in general, higher than online surveys.[62 63] One potential reason is rising concerns about security and privacy in the digital age.[64 65] Indeed, common barriers to engaging in internet-administered interventions include security and confidentiality concerns[19] Possibly, parents may have experienced opt-out via the post as implying higher levels of security and privacy than using an electronic personal identification method (BankID) to opt-out online. However, the utilisation of different modes of opt-out was easy to implement and given the variation in response rate across the four opt-out modes, future studies may wish to offer potential participants multiple modes of opt-out. Further, future studies could examine additional potential modes of opt-out, for example, via short message service.[66]

The trend for parents with an upper secondary school education to opt out of the study while parents with a tertiary education chose to participate is supported by the wider literature. For example, parents of adolescents with psychosocial difficulties who have lower levels of education perceive more barriers towards accessing psychosocial care, than parents with university-level education.[67] Moreover, evidence suggests online survey responders tend to be more highly educated than non-responders[68] and adults with lower levels of education struggle to find health information online[69 70] and require more guidance to use internet-administered interventions.[70] Higher education levels are also associated with increased effectiveness and higher completion rates of guided internet-administered CBT interventions in a general adult population.[71] Future research is needed to improve the acceptability and feasibility of internet-administered interventions for people with lower education levels.

In line with previous research with depressed adult populations,[72 73] the most common reason for non-participation was not perceiving a need for psychological support. Similar findings were reported in psychological intervention studies for adult patient with cancer/caregiver dyads[3] and informal caregivers of stroke survivors.[57] Around 14%–30% of parents of children treated for cancer report clinically relevant levels of mental health

difficulties, such as depression, anxiety and post-traumatic stress.[31 74 75] Therefore, at a group level, a majority of parents report levels of psychological distress within a normal range[36] and it can be anticipated most do not experience a need for psychological support. However, as the 11.8% consent rate is under the prevalence rate of psychological distress in the population, some parents in need of psychological support did not consent and research to further increase participation among this group is warranted.

Study invitation timing was the most commonly cited reason for non-participation in response to the open question, corroborating findings of a similar study conducted with parents of children with cancer examining potential barriers to recruitment.[29] Parents found it difficult to identify the most appropriate time point for intervention, as periods of increased distress may differ between parents.[29] Other research has found invitation timing plays a significant role in a person's decision to participate in studies of psychosocial interventions for couples coping with cancer[76] and that it is difficult to find an optimal time point to recruit patient with cancer/caregiver dyads into intervention research.[77] A potential solution may be to not place any time limit from diagnosis/end of treatment.

Finally, lack of interest in internet-administered self-help was a common reason for non-participation. Despite well-demonstrated clinical effectiveness of e-MH interventions,[78] difficulties with recruitment into trials are common[79] and rates of implementation are low.[80] Barriers include lack of trust, poor digital health literacy, low levels of confidence in using digital technology, and preferences for face-to-face interventions.[17–20] Research indicates acceptability may increase when potential users are provided with information about intervention content and/or effectiveness[18 81] and future research may wish to examine the provision of additional intervention information, including, for example, screenshots or videos, on recruitment rates.

### Ethical considerations
One concern around an opt-out strategy is the use of repeated reminders/contact attempts, and potentially coercing participants.[21] While research ethics committees often require researchers to use opt-in recruitment strategies to protect participant confidentiality and (medical) privacy,[82] this minimises the opportunity for researchers to speak with potential participants directly to explain the study.[83] Consistent with other research using opt-out recruitment whereby few concerns have been raised,[82] only one complaint concerning reminder calls was received during the study, indicating an opt-out recruitment strategy is acceptable to the population.

### Strengths and limitations
This study has several strengths. First, parents were identified and randomly invited into the study via the Swedish Childhood Cancer Registry and a population registry

held by the Swedish Tax Agency (NAVET). As such, invited parents are a nationally representative sample of parents of children treated for cancer in Sweden. Second, parents were provided with different modes of opt-out (eg, post, online, telephone or email). To the best of our knowledge, this is the first study to compare opt-out rates across different opt-out modalities. Third, by using both closed and open questions to examine reasons for non-participation, results provide us with a broad understanding of reasons for non-participation which may inform the development of future recruitment strategies. The study also has limitations. First, it was not designed to compare recruitment rates across different recruitment strategies and conclusions on the effectiveness of the opt-out strategy versus other recruitment strategies cannot be drawn. Future studies may look to investigate the effect of an opt-out recruitment strategy with an opt-in recruitment strategy on recruitment rates. Second, we did not collect sociodemographic variables of non-responders and therefore it was not possible to assess whether the sample is representative of the overall population in relation to key sociodemographic variables. Some sociodemographic data (eg, sex, age, geographical location) could have been obtained for all potential participants via the NAVET registry, which would have enabled comparisons between participants who consented, opted out, and were non-responders[84] however we did not include the collection of this data in the ethics application. Third, while the open question regarding reasons for non-participation allowed for a broad understanding of reasons for non-participation, embedding semistructured interviews to explore reasons for non-participation in more depth may have facilitated a richer understanding of reasons for non-participation in the population.

## CONCLUSIONS

This is the first study to examine the use of an opt-out strategy and different modes of opt-out within a feasibility trial examining the feasibility and acceptability of an internet-administered, guided, LICBT-based self-help intervention for parents of children treated for cancer. Though recruitment into clinical trials is challenging, projected recruitment rates were exceeded. Despite the most frequent mode of opt-out being paper via the post, given the implementation of different modes of opt-out was straightforward, coupled with the variation in response rate across the four opt-out modes, future studies should consider using multiple modes of opt-out. Developing a more comprehensive understanding of reasons for non-participation may help inform planning and decision-making concerning study design, and potentially enhance recruitment rates and reduce research waste in future trials of internet-administered interventions for both parent/caregiver and wider populations. Future research may wish to compare opt-out versus opt-in strategies within an RCT and embed qualitative interviews to explore reasons for non-participation

and develop a more in-depth understanding of potential barriers to trial participation.

**Author affiliations**
[1]Healthcare Sciences and e-Health, Department of Women's and Children's Health, Uppsala University, Uppsala, Sweden
[2]Department of Paediatrics and Adolescent Medicine, TYKS Turku University Hospital, Turku, Finland
[3]Pediatric Oncology and Pediatric Surgery, Department of Women's and Children's Health, Karolinska Institute, Stockholm, Sweden

**Acknowledgements** The authors thank the four PRPs (Helena Börjesson, Martin Hedqvist, Anna Mautner and Erik Olsson) for their feedback on study invitation material. In addition, we wish to thank Fabian Holmberg, Ian Horne and Ylva Hägg Sylvén (Portal team members) for their assistance with developing the opt-out procedure on the Portal. We also thank research assistant, Christina Reuter, for assisting with some elements of data preparation and analysis.

**Contributors** Author contributions are written in accordance with the CRediT statement: JH—formal analysis, investigation, data curation, writing (original draft), visualisation. JW—Methodology, formal analysis, writing (original draft), supervision, project administration. AvE—validation, investigation, data curation, formal analysis, writing (review and editing). PL—resources, writing (review and editing). LvE—guarantor, conceptualisation, methodology, resources, writing (review and editing), supervision, project administration, funding acquisition.

**Funding** This work is supported by the Swedish Research Council (grant number K2015-99X-20836-08-4 and 2018-02578), the Swedish Cancer Society (grant number 150673 and 180589), the Swedish Childhood Cancer Foundation (grant number PR2017-0005), and funding via the Swedish Research Council to U-CARE, a Strategic Research environment (Dnr 2009-1093).

**Disclaimer** The funders had no role in study design, data collection and analysis, decision to publish or preparation of the manuscript.

**Competing interests** None declared.

**Patient and public involvement** Patients and/or the public were involved in the design, or conduct, or reporting, or dissemination plans of this research. Refer to the Methods section for further details.

**Patient consent for publication** Not required.

**Ethics approval** This study involves human participants and was approved by the Regional Ethical Review Board in Uppsala, Sweden (Dnr: 2017/527), and was conducted in accordance with the Helsinki Declaration and Good Clinical Practice (GCP) guidelines. Participants gave informed consent to participate in the study before taking part.

**Provenance and peer review** Not commissioned; externally peer reviewed.

**Data availability statement** Data are available in a public, open access repository. The research data supporting the findings of this study are stored in Zenodo repository with the identifier https://doi.org/10.5281/zenodo.6136920. Access to the data stored in Zenodo is available upon written request from the corresponding author (LvE).

**ORCID iD**
Joanne Woodford http://orcid.org/0000-0001-5062-6798

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
