## [Reviewer comments · BMJ Open]

ARTICLE DETAILS

TITLE (PROVISIONAL)	Opt-out rates and reasons for non-participation in a single-arm feasibility trial (ENGAGE) of a guided internet-administered CBT-based intervention for parents of children treated for cancer: a nested cross-sectional survey
AUTHORS	Hagström, Josefin; Woodford, Joanne; von Essen, Agnes; Lähteenmäki, Paivi; von Essen, Louise

VERSION 1 – REVIEW

REVIEWER	Northgraves, Matthew Hull York Medical School
REVIEW RETURNED	03-Jan-2022

GENERAL COMMENTS	Thank you for the opportunity to review this paper looking at the use of different methods of opt-out. Whilst not providing a comparison with the use of opt-in would have been interesting, it is acknowledged that including comparisons in study designs may not always be possible. The paper provides information around reasons of non-participation that could be used to inform future research. A couple of minor comments regarding the abstract and methods are below: Abstract Rather than start by going straight into the objective, my personal thoughts are the abstract should begin with a very brief background to the study even if just a single sentence like the examples below: https://bmjopen.bmj.com/content/11/12/e050051 https://bmjopen.bmj.com/content/11/12/e050004 Introduction Good background and rational to the study provided. Methods Repeatable methods describe, would be interesting to know whether the use of personalised invitation letters effects the number of non-responders. e.g. fewer opt-out or more respond to opt-out without the need subsequent reminders before opting out. Results and discussion Results clearly presented and reflected in the discussion. Identifies limitations for the study and suggests future research direction.
--

REVIEWER	Liu, Na The Second Affiliated Hospital of Chongqing Medical University
REVIEW RETURNED	07-Feb-2022

GENERAL COMMENTS	Thank you very much for inviting me to make suggestions on this manuscript. The manuscript research design is rigorous, logical, accurate and comprehensive in analyzing the reasons for participants' participation and withdrawal. However, the primary objective of this paper is opt-out modes, but the results and discussion of the paper lack further explanation of the differences between different opt-out modes and the potential significance of choosing different opt-out modes. What practical guiding significance does this study have for future research? Please explain further.
---

VERSION 1 – AUTHOR RESPONSE

Reviewer: 1
Dr. Matthew Northgraves, Hull York Medical School

Comments to the Author:

Thank you for the opportunity to review this paper looking at the use of different methods of opt-out. Whilst not providing a comparison with the use of opt-in would have been interesting, it is acknowledged that including comparisons in study designs may not always be possible. The paper provides information around reasons of non-participation that could be used to inform future research. A couple of minor comments regarding the abstract and methods are below:

2. Abstract

Rather than start by going straight into the objective, my personal thoughts are the abstract should begin with a very brief background to the study even if just a single sentence like the examples below:

**<https://bmjopen.bmj.com/content/11/12/e050051>
<https://bmjopen.bmj.com/content/11/12/e050004>**

Thank-you for this recommendation. We have added a very brief background to the Abstract (see page 2):

Difficulties with recruitment into clinical trials are common. An opt-out recruitment strategy, whereby potential participants can decline further contact about a study (opt-out), and non-responders are contacted, may facilitate participation.

3. Introduction

Good background and rational to the study provided.

Thank-you for this feedback.

4. Methods

Repeatable methods describe, would be interesting to know whether the use of personalized invitation letters effects the number of non-responders. e.g. fewer opt-out or more respond to opt-out without the need subsequent reminders before opting out.

Thank-you for this comment. We have submitted the study-within-a-trial (SWAT) as a separate manuscript, have recently received reviewers' comments and the manuscript has been revised and resubmitted, and we hope will be accepted for publication soon. One of the secondary outcomes was the proportion of potential participants invited into the trial in each group that opted out of the ENGAGE host feasibility trial. The number for opting out of the trial was smaller in the personalized invitation letter group than the non-personalized invitation letter group. However, we would prefer to only report this data in the SWAT manuscript and have informed the journal that this data is not presented for publication elsewhere. As such, it is not possible to add this data to the manuscript.

Results and discussion

Results clearly presented and reflected in the discussion. Identifies limitations for the study and suggests future research direction.

Thank-you for this comment.

Reviewer: 2

Miss Na Liu, The Second Affiliated Hospital of Chongqing Medical University

Comments to the Author:

Thank you very much for inviting me to make suggestions on this manuscript. The manuscript research design is rigorous, logical, accurate and comprehensive in analyzing the reasons for participants' participation and withdrawal. However, the primary objective of this paper is opt-out modes, but the results and discussion of the paper lack further explanation of the differences between different opt-out modes and the potential significance of choosing different opt-out modes. What practical guiding significance does this study have for future research? Please explain further.

Thank-you for your positive comments on the manuscript. We have four primary objectives, to examine opt-out and consent rates, mode and time point of opt-out, and sociodemographic characteristics of those who opted out versus chose to participate in the ENGAGE feasibility trial. At present, we have discussed findings relating to each of these objectives. We have made some minor amendments to the wording of the study objectives, to make the four primary objectives clear (see page 5):

Primary objectives examined: (1) opt-out and consent rates; (2) mode of opt-out; (3) time point of opt out; and (4) sociodemographic characteristics of those who opted out versus chose to participate in the ENGAGE feasibility trial.

However, examining different modes of opt-out is a novel feature of the manuscript and we have elaborated this in the Discussion (see page 17):

To the best of our knowledge, this is the first study to specifically examine the rate of opt-out across four different modes of opt-out. Commonly, studies utilising opt-out recruitment strategies include only one mode of opt-out, for example via the post [26; 58] or telephone.[59] The most frequent mode of opt-out in the ENGAGE feasibility trial was paper via the post (47.6%), followed by telephone (32.3%), with findings similar to a quantitative telephone survey with military veterans examining rural and urban differences in attitudes towards mental health care and influence on mental health service use, whereby opt-out via the post was more frequent (75%) than opt-out via the telephone (25%).[49] Interestingly, only 14.6% of parents opted out online via the Portal in the ENGAGE feasibility trial. In comparison with the general adult population, parents of a child treated for cancer three months to five years following end of treatment may be assumed to be younger, with good digital knowledge and skills.[60] Therefore it may have been anticipated that parents would prefer opt-out online. However, systematic reviews and meta-analyses have shown postal survey response to be, in general, higher than online surveys.[61,62] One potential reason is rising concerns about security and privacy in the digital age.[63,64] Indeed, common barriers to engaging in internet-administered interventions include security and confidentiality concerns [19] Possibly, parents may have experienced opt-out via the post as implying higher levels of security and privacy than using an electronic personal identification method (BankID) to opt-out online. However, the utilisation of different

modes of opt-out was easy to implement and given the variation in response rate across the four opt-out modes, future studies may wish to offer potential participants multiple modes of opt-out. Further, future studies could examine additional potential modes of opt-out, for example, via short message service (SMS).[65]

We have also added further amendments to the Conclusion section regarding future research (see page 19):

This is the first study to examine the use of an opt-out strategy and different modes of opt-out within a feasibility trial examining the feasibility and acceptability of an internet-administered, guided, LICBT based self-help intervention for parents of children treated for cancer. Though recruitment into clinical trials is challenging, projected recruitment rates were exceeded. Despite the most frequent mode of opt-out being paper via the post, given the implementation of different modes of opt-out was straightforward, coupled with the variation in response rate across the four opt-out modes, future studies should consider using multiple modes of opt-out.